# Serological Evidence of Crimean–Congo Haemorrhagic Fever in Livestock in the Omaheke Region of Namibia

**DOI:** 10.3390/microorganisms12040838

**Published:** 2024-04-22

**Authors:** Alaster Samkange, Pricilla Mbiri, Ophelia Chuma Matomola, Georgina Zaire, Anna Homateni, Elifas Junias, Israel Kaatura, Siegfried Khaiseb, Simson Ekandjo, Johannes Shoopala, Magrecia Hausiku, Albertina Shilongo, Mushabati Linus Mujiwa, Klaas Dietze, Frank Busch, Christian Winter, Carolina Matos, Sabrina Weiss, Simbarashe Chitanga

**Affiliations:** 1School of Veterinary Medicine, Faculty of Health Sciences & Veterinary Medicine, University of Namibia, Private Bag 13301, Windhoek 10005, Namibia; alastersamkange@gmail.com (A.S.); pmbiri@unam.na (P.M.); cmatomola@unam.na (O.C.M.); ikaatura@unam.na (I.K.); khaisebs@gmail.com (S.K.); mmushabati@unam.na (M.L.M.); 2Directorate of Veterinary Services, Ministry of Agriculture, Water and Land Reform, Private Bag 13184, Windhoek 10005, Namibia; georgina.zaire@mawlr.gov.na (G.Z.); annahomateni@gmail.com (A.H.); elifasjunias@gmail.com (E.J.); s.ekandjo@yahoo.com (S.E.); johannes.shoopala@mawlr.gov.na (J.S.); magrecia.hausiku@mawlr.gov.na (M.H.); albertina.shilongo@mawlr.gov.na (A.S.); 3Institute of International Animal Health/One Health, Friedrich-Loeffler Institute, 17489 Greifswald, Germany; klaas.dietze@fli.de (K.D.); frank.busch@fli.de (F.B.); 4Centre for International Health Protection, Robert Koch Institute, 13353 Berlin, Germany; winterc@rki.de (C.W.); matosc@rki.de (C.M.); weisss@rki.de (S.W.); 5Department of Biomedical Sciences, School of Health Sciences, University of Zambia, P.O. Box 50110, Lusaka 10101, Zambia

**Keywords:** Crimean–Congo haemorrhagic fever, seroprevalence, sheep, cattle, Omaheke, Namibia, tick-borne

## Abstract

This research examined the positivity ratio of Crimean–Congo haemorrhagic fever (CCHF) antibodies in cattle and sheep within Namibia’s Omaheke region after a human disease outbreak in the same geographical area. A total of 200 samples (100 cattle and 100 sheep) were randomly collected from animals brought to two regional auction sites, and then tested using the ID Screen^®^ CCHF Double Antigen Multi-Species Enzyme-Linked Immunosorbent Assay kit. Of the cattle samples, 36% tested positive, while 22% of the sheep samples were seropositive. The cattle had a significantly higher positivity ratio than sheep at the individual animal level (*p* = 0.0291). At the herd level, 62.5% of cattle herds and 45.5% of sheep flocks had at least one positive animal, but this difference was statistically insignificant (*p* = 0.2475). The fourteen cattle farms with at least one seropositive animal were dispersed across the Omaheke region. In contrast, the ten sheep farms with seropositive cases were predominantly situated in the southern half of the region. The study concluded that the CCHF is endemic in the Omaheke region and likely in most of Namibia, underscoring the importance of continued surveillance and preventive measures to mitigate the impact of CCHFV on animal health and potential spillover into human populations.

## 1. Introduction

The Crimean–Congo Hemorrhagic Fever Virus (CCHFV), belonging to the genus *Orthnairovirus* within the order *Bunyavirales* and the family *Nairoviridae*, is an emerging zoonotic virus transmitted by ticks. It is considered endemic in certain regions of Africa, Asia, and Europe [1]. The geographical range of CCHFV closely correlates with the distribution of *Hyalomma* ticks, primarily *Hyalomma marginatum* and *Hyalomma rufipes*, the main reservoirs and vectors responsible for transmitting the virus to humans [2,3]. Crimean–Congo haemorrhagic fever (CCHF) is considered one of the diseases with significant outbreak, epidemic, and pandemic potential within Africa [4], and this is coupled with its high fatality rate, various transmission routes and challenges associated with its diagnosis, prevention, and treatment (www.who.int/news-room/fact-sheets/detail/crimean-congo-haemorrhagic-fever, accessed on 30 August 2023). The virus typically circulates in an enzootic cycle involving animals and ticks, with the ticks serving as both vectors and reservoirs of the virus [5]. Whilst *Hyalomma marginatum* is considered the principal species involved in the transmission and maintenance of the virus, the role of other tick species is still undefined, even though molecular evidence of the presence of the virus in several other tick species exists [2,5]. Maintenance in the principal tick vector has been shown to occur through tick-to-tick transmission and this has been shown to happen transovarially, transstadially, sexually and/or during co-feeding [2], ensuring the persistence of the virus in the tick and establishment in an area once introduced.

Whilst non-human vertebrates have been shown not to suffer clinical disease, they have a short phase in which they are viraemic, thus becoming essential amplifiers of infection as they can transmit the virus to ticks during that phase [6]. Humans serve as accidental hosts and are the exclusive species in nature known to exhibit clinical effects from CCHFV [7], with transmission known to occur through the bite of an infected tick, as well as through direct contact with the blood or tissues of viraemic animals or human patients carrying the CCHF virus [1]. Approximately 90% of human infections either show no symptoms or result in a mild, non-specific fever without additional clinical consequences [8]. The non-fatal cases can be characterised by a rapid appearance of fever, severe headache, and overall discomfort. The infrequent fatal cases are a result of extensive bleeding, hypovolemic shock, and multi-organ failure [9]. 

Between 2016 and 2023, Namibia recorded six outbreaks of CCHF in humans, with the latest one being declared in May 2023 [10,11]. Despite the country being considered endemic, little is known about the epidemiology and/or epizootiology of CCHFV in the country. An early warning system for CCHF is crucial for preventing outbreaks and reducing the effects of a disease outbreak [12], making it critical for surveillance data to inform disease epidemiology to be available. The usefulness of domestic livestock as sentinels to indicate the circulation of CCHFV in any geographic location has long been established [7], and it was with this in mind that this study was conducted to estimate the spread of the CCHFV by screening livestock from different farms in the Omaheke region of Namibia.

## 2. Materials and Methods

### 2.1. Study Sites

The study area was the Omaheke region of Namibia, which is located in the central-eastern area of the country (Figure 1). The study site was purposefully chosen as it is the region where CCHF outbreaks were reported in 2016 and, more importantly, in 2023 [10].

### 2.2. Study and Sampling Design

Blood samples were collected from animals presented for auction at two locations in the Omaheke region of Namibia—specifically, Gobabis for cattle (24 farms) and Leonardville for sheep (22 farms) (Appendix A). Auctions were chosen because they allowed for the sampling of animals from wider geographical areas covered by the auction catchment area. Samples were collected on a single day at each auction site in June 2023. A simple random sampling method was used to select the sampled animals at the auction facility. Before the auction, all animals from various farms were logged into the computer system and allocated sequential identification numbers. The 2013 version of Microsoft Excel was utilized to generate random numbers, aligning with the predetermined sample size for each animal species at the two auctions. The collected blood samples were centrifuged at 25,000× *g* for five minutes, and the sera were collected into 2 mL sterile cryovials and stored at −20 °C until testing.

### 2.3. Enzyme–Linked Immunosorbent Assay (ELISA)

The ID Screen^®^ CCHF Double Antigen Multi-Species Enzyme-Linked Immunosorbent Assay (IDvet, Grabels, France) was used to test all 200 of the serum samples for antibodies against CCHFV. This ELISA test kit is a double antigen kit that measures IgM and IgG antibodies and has a sensitivity and specificity of 98.9% and 100%, respectively [13]. All the reagents were brought to room temperature (20–26 °C) and then homogenised by vortexing. The assay was then performed, and the results were validated according to the manufacturer’s instructions. A test run was deemed valid if it fulfilled the following conditions: the mean optical density of the positive control (ODpc) exceeded 0.35, and the ratio of the mean ODpc to the mean optical density of the negative control (ODnc) was above 3. The interpretation of the test’s optical densities relied on the ratio between the mean optical density of the samples and ODpc, presented as a percentage (S/p × 100). Samples with an S/p% equal to or less than 30% were categorised as negative, whereas samples with an S/p% greater than 30% were considered positive. The test results were read at 450 nm using an ELISA reader.

### 2.4. Data Analysis

Data were captured in a Microsoft Excel^®^ spreadsheet (version 2013) and analysed using descriptive statistics and the *z*-test to compare proportions (https://epitools.ausvet.com.au/, accessed on 7 October 2023). A *p*-value of <0.05 was considered significant. Geospatial data visualisation was performed using ArcGIS ArcMap 10.8.1 (ESRI, Redlands, CA, USA) with GIS data from https://gadm.org/data.html

## 3. Results

In total, 200 animals were sampled (100 cattle and 100 sheep). Most tested animals originated from different areas within the Omaheke region. The sampled cattle came from 24 different farms and the sampled sheep were from 22 different farms. 

The overall positivity ratio from the anti-CCHFV ELISA was 29% (58/200), with the findings segregated as follows: cattle—36% (36/100) and sheep—22% (22/100) (Appendix A). At the individual animal level, cattle had significantly higher seropositivity than sheep (*p* = 0.0291). The overall farm-level positivity ratio was 54.3% (25/46), whilst by animal species, the cattle farm positivity ratio was 62.5% (15/24), and the sheep farm positivity ratio was 45.5% (10/22), but this difference was statistically insignificant (*p* = 0.2475). 

An analysis of the spatial distribution of positive cases revealed that the fifteen cattle farms, each with at least one seropositive animal, were dispersed across the Omaheke region, except for one farm located in the neighbouring Khomas region (Figure 1). The ten seropositive sheep farms were primarily located in the southern half of the Omaheke region, with one farm situated in the adjacent Hardap region to the south. 

## 4. Discussion

Our study not only provides the first evidence of the circulation of CCHFV in livestock in Namibia, but it also shows high seropositivity in cattle and sheep. This indicates that there is risk of human infection with CCHFV in Omaheke, thus explaining the sporadic outbreaks on infection reported in humans in the country [10]. 

Globally and regionally, the reported prevalences of seropositivity to CCHF vary widely [14,15,16,17,18]. Within the southern African region, seroepidemiological studies to screen for CCHFV exposure in animals have been conducted in Zambia [17,18], South Africa [16,19,20], Malawi [15], and Zimbabwe [19,20]. Across these countries, the reported seroprevalence has varied widely, and the differences in the animal species sampled in the different studies could explain this difference as all animal species have different susceptibilities to infection. Cattle, which have been predominantly sampled in regional studies, are reportedly more at risk of infection with CCHFV than small stock and thus generally tend to have higher prevalences [14]. 

In our study, the positivity ratio in cattle was higher than that of sheep. This is in line with reports from other countries where cattle and other species have been sampled together and their prevalence compared, with cattle consistently having a higher prevalence [21,22]. However, some studies have reported a relatively higher prevalence in sheep compared to cattle [23]. There are several factors which have been identified to influence seroprevalence between studies, and these include the serological test of choice, host differences (age, sex, and breed), livestock management systems, climatic conditions as well as vector abundance and competence [17,24,25].

Analysis of the spatial distribution of CCHFV-seropositive animals showed widespread distribution across the region. Whilst *Hyalomma* species are traditionally considered the principal vector for the virus [3], there is evidence of other tick species’ involvement in the virus’ maintenance and transmission [2]. The relative importance of different tick species (besides *Hyalomma*) in the transmission of CCHFV is further evidenced by findings from a study by Phonera et al. [15], in which they found the highest seroprevalence of CCHFV in regions in which there were no previous reports of *Hyalomma* presence. Considering that ticks are widely spread in Namibia [26,27,28,29], it is not surprising that the virus seems to be widely distributed in the study area. This indicates that there is risk of infection in humans in the region, especially considering the predominance of livestock farming, a factor which is regarded as a risk factor [30].

## 5. Conclusions

The current study conducted in the Omaheke region, following a reported human case of CCHF, highlights the widespread exposure of CCHFV in the region, indicating the existence of the threat of further human infections. Thus, consistent surveillance is needed in the region and possibly the whole country, as well as enhanced awareness amongst the community members, especially the risk groups. Such awareness measures include education on CCHFV transmission and various prevention measures which can be implemented at the individual level. Considering the serological evidence of CCHFV circulation in animals, it is important to conduct serosurveys in humans and molecular screening in ticks to develop a more comprehensive understanding of CCHFV epidemiology in Namibia. 

## Figures and Tables

**Figure 1 microorganisms-12-00838-f001:**
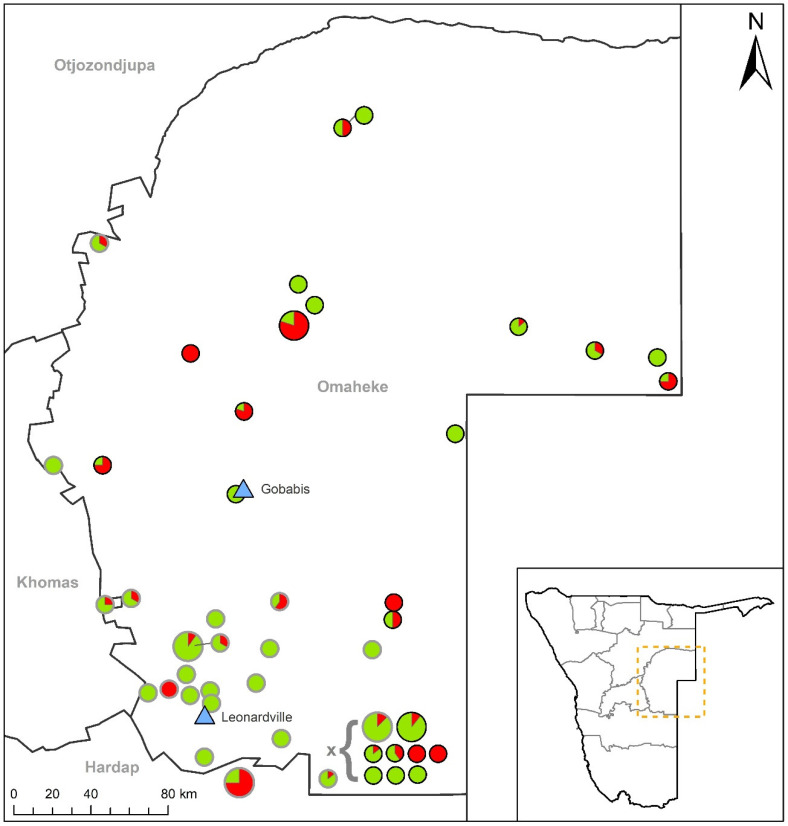
Map of the study area of Omaheke and the adjacent regions in Namibia. Blue triangles depict the auction centres where sampling took place. Black-ringed circles depict cattle farms, grey-ringed circles depict sheep farms. Green and red filling of circles depict the share of negative and positive samples, respectively. Small circles indicate a total number of tested animals < 10, large circles indicate the number of tested animals to be 10 and more.

## Data Availability

The raw data generated from this study are not available due to privacy reasons.

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
