# Peer review of "Serological Evidence of Crimean–Congo Haemorrhagic Fever in Livestock in the Omaheke Region of Namibia"

_microorganisms, 2024, doi:10.3390/microorganisms12040838_

Round 1

Reviewer 1 Report

Comments and Suggestions for Authors

The authors present a brief communication of a serosurvey of cattle and sheep from one region in Namibia. The study sampled animals brought to auction, and ages, sexes, presence/absence of ticks, were not measured. The authors state that sampling was “random” but should explain this a bit more, as random sampling might preclude the per-farm analysis. The authors could also mitigate this by providing a map and/or a table that states the locations or sampling numbers on a per-farm basis. Or remove the per-farm analysis/discussion and “analysis of spatial distribution”. Overall, the study provides simple estimates of seropositivity in two types of livestock from one region of Namibia. These data are important for public & veterinary health authorities, as well as those studying CCHFV global distribution and ecology.

Below are some specific comments and suggestions that the authors should address. Some of which may be repeated from the summary above.

Three major issues:

Lines 83, 89-91, 128, and especially line 155. We need a map. If it is not permitted to have a figure for a Communication, then please provide one as a supplemental. Include to location of the two auctions within Omaheke, locations of farms from which the samples were taken – ideally indicating sample size if possible, but certainly indicating positivity in binary. Also include an inset showing the location within a larger geographic scale.

Methods: the authors must state the time period in which the samples were obtained. Was this a follow-up investigation to the May 2023 nosocomial outbreak? This is stated only in the conclusion (line 167), but it is essential to report the timing in the methods. As the authors, themselves state, transmission activity can vary widely and can be quite focal/clustered over small spatiotemporal scales (farms or regions as reported here). Knowing that this was a follow-up to a single human case that resulted in a large outbreak would be important to know/consider from the beginning of the manuscript.

Finally, two points about approach, study design and statistics: First, Line 120-121 –it is important to state briefly how many per farm. Were animals sampled randomly per farm? how was “random” sampling handled? Line 92-93 – state briefly what the random method was…a per-animal coin toss? Or was it more systematic (e.g., three “random” animals per farm)?

There could be some structure to the data that truly random sampling wouldn’t cover, particularly within-farm error. The probability of detecting a case on a positive farm increases with increased sampling. But also, as the authors point out, that if one animal on a farm is sero-positive, it is highly likely that other will also be sero-positive. The authors analyze data on a per-farm basis (line 125-126) and report a p-value for this comparison in the abstract (line 33) but not in the results – if they can justify why a per-farm analysis was possible using this sampling method, then they should report all results in the results.

A final comment on this issue: I understand that this is a small study and the authors do not wish to extend the analysis too far. I just have doubts about presenting the per-farm analysis and there are some limitations that the authors should address about the conclusions that can be inferred about the "geographic distribution" in the region. So, instead of changing the presentation/analysis of the study and data, I suggest that the best way to clear this up is to provide a supplemental table with sampling information: farm A, location (region?), N cattle, n/N seropositive (percent);

farm B, location X sheep, x/X seropositive (percent);… etc.

And let the reader make their own assessments of the data.

Minor comments/suggestions:

Line 33 – “…was not statistically significant” but the authors should also report this finding in the results…

Line 45 – CCHFV is not endemic America

Line 52 – the web address should begin with “(www.who.int/news-room...”

Line 59 – “sustenance” means “food or nourishment”. I suggest re-writing, “These factors ensure that the virus is sustained in the tick [population] and becomes established in an area once introduced.” or something. Consider completely revising the sentence, as it is not clear what the intention/importance of the sentence should be.

Line 63 – clarify “…transmit the virus to naïve ticks…”

Line 64 – perhaps it is important to clarify that these are the only non-laboratory animals (or maybe “in nature” or something) to exhibit clinical effects, as laboratory models of pathogenicity (transgenic mice and cynomolgus macaques) certainly exist (see citation [8].)

Line 65 – remove “crushing of an infected tick”. This is often cited as a transmission route but there is simply no evidence supporting this. It certainly is not mentioned in the cited reference [1].

Line 74 – epidemiology (in humans)? Or epizootiology (spillover)? or do they mean the enzootic transmission ecology of CCHFV?

Beginning line 122, results: “ratios” are reported. “Rates” imply there is a time component.

Line 155 – where is the analysis of the spatial distribution of CCHFV seropositive animals?

Comments on the Quality of English Language

Very very minor editing of mostly typos and one or two incorrect uses of words (e.g., "sustenance" in Line 59).

Reviewer 2 Report

Comments and Suggestions for Authors

The manuscript revealed the serologic prevalence of CCHFV infection in cattle and sheep in the Omaheke region where the multiple outbreaks of CCHFV have occurred. The aim of this study is important to understand the spread of the virus in this region. However, the authors only showed detection data of IgG and IgM by ELISA. They should also show details of the samples used in this study, such as where and when the samples were collected, age and sex of the cattle, geographical relationship between the location of the farms and the location of the outbreak. You can discuss more about how the virus spread through the scientific evidence.
 I believe that at least these data and discussions will support the possibility of publishing this manuscript.

Reviewer 3 Report

Comments and Suggestions for Authors

The relevance of the work is determined by the severity of CCHF and the possibility of nosocomial outbreaks. Preventive measures, including equipping health care facilities with diagnostic kits and training of health care personnel, are important to reduce morbidity and mortality from this infection. Data on seroprevalence in domestic animals provide the data necessary to assess the risk of spread. In general, the communication is well written and clear, but still there are not enough GPS coordinates of the farms where the samples were collected. An important conclusion that follows from the article, that data for cattle is more indicative than for sheep, follows from the location of the farms, which we can only glean from knowing the coordinates. It might be worthwhile to shorten the general words in the introduction, but still provide these data, as well as suspected species of ticks in Namibia that could be a vector for the virus other than Hyalomma marginatum

Round 2

Reviewer 2 Report

Comments and Suggestions for Authors

The authors had well improved the manuscript following the comments.